# Nanoparticles for Thrombolytic Therapy in Ischemic Stroke: A Systematic Review and Meta-Analysis of Preclinical Studies

**DOI:** 10.3390/pharmaceutics17020208

**Published:** 2025-02-06

**Authors:** Jesús Prego-Domínguez, Fernando Laso-García, Nuria Palomar-Alonso, María Pérez-Mato, Esteban López-Arias, Antonio Dopico-López, Pablo Hervella, María Gutiérrez-Fernández, María Alonso de Leciñana, Ester Polo, Beatriz Pelaz, Pablo del Pino, Francisco Campos, Clara Correa-Paz

**Affiliations:** 1Head of Epidemiologic Surveillance Service, Public Health General Directorate, Consellería de Sanidade, 15703 Santiago de Compostela, Spain; jesus.prego.dominguez@sergas.es; 2Neurological Sciences and Cerebrovascular Research Laboratory, Department of Neurology and Stroke Centre, Neurology and Cerebrovascular Disease Group, Neuroscience Area La Paz Institute for Health Research–idiPAZ, La Paz University Hospital-Universidad Autónoma de Madrid, 28049 Madrid, Spain; fernilaso.9@gmail.com (F.L.-G.); mariagutierrezfdez@hotmail.com (M.G.-F.); malecinanacases@salud.madrid.org (M.A.d.L.); 3Translational Stroke Laboratory (TREAT), Clinical Neurosciences Research Laboratory (LINC), Health Research Institute of Santiago de Compostela (IDIS), 15706 Santiago de Compostela, Spain; nuria.palomar.alonso@sergas.es (N.P.-A.); maria.perez.mato@sergas.es (M.P.-M.); esteban.lopez.arias@sergas.es (E.L.-A.); antonio.dopico.lopez@sergas.es (A.D.-L.); 4Neuroimaging and Biotechnology Laboratory (NOBEL), Clinical Neurosciences Research Laboratory (LINC), Health Research Institute of Santiago de Compostela (IDIS), Hospital Clínico Universitario, Rúa Travesa da Choupana s/n, 15706 Santiago de Compostela, Spain; pablo.hervella.lorenzo@sergas.es; 5Center for Research in Biological Chemistry and Molecular Materials (CiQUS), University of Santiago de Compostela (USC), 15705 Santiago de Compostela, Spain; ester.polo@usc.es (E.P.); beatriz.pelaz@usc.es (B.P.); pablo.delpino@usc.es (P.d.P.)

**Keywords:** cell membrane-derived nanomedicines, ischemic stroke, meta-analysis, nanoparticles, rtPA, thrombolysis

## Abstract

**Background:** Recombinant tissue plasminogen activator (rtPA) remains the standard thrombolytic treatment for ischemic stroke. Different types of nanoparticles have emerged as promising tools to improve the benefits and decrease the drawbacks of this therapy. Among them, cell membrane-derived (CMD) nanomedicines have gained special interest due to their capability to increase the half-life of particles in blood, biocompatibility, and thrombus targeting. In order to update and evaluate the efficacy of these nanosystems, we performed a meta-analysis of the selected in vivo preclinical studies. **Methods:** Preclinical in vivo studies in ischemic stroke models have been identified through a search in the Pubmed database. We included studies of rtPA-nanoparticles, which assessed infarct volume and/or neurological improvement. Nanosystems were compared with free (non-encapsulated) rtPA treatment. Standardized mean differences were computed and pooled to estimate effect sizes for lesion volumes and neurological scores. Subgroup analyses by the risk of bias, type of nanoparticle, and time of administration were also performed. **Results:** A total of 18 publications were included in the meta-analysis. This was based on defined search inclusion criteria. Our analysis revealed that rtPA-nanoparticles improved both lesion volume and neurological scores compared with the free rtPA treatment. Moreover, CMD nanomedicines showed better evolution of infarct volume compared to the other nanoparticles. Funnel plots of lesion volume exhibited asymmetry and publication bias. Heterogeneity was generally high, and the funnel plot and Egger test showed some evidence of publication bias that did not achieve statistical significance in the trim-and-fill analysis. **Conclusions:** rtPA-encapsulating nanosystems were shown to decrease infarct volume and improve neurological scales compared to the standard treatment, and CMD nanomedicines had the greatest beneficial effect.

## 1. Introduction

Stroke is a neurovascular pathology that occurs unexpectedly and has a disastrous outcome. Approximately 15 million people experience a stroke episode every year, becoming one of the leading causes of disability and death worldwide. Ischemic stroke is the result of a sudden interruption of the cerebral blood flow (CBF) usually caused by the occlusion of a cerebral artery by a blood clot [1].

Rapid restoration of blood flow either through pharmacological thrombolysis or mechanical thrombectomy increases significantly the probability of a good outcome [2]. The recombinant tissue plasminogen activator (rtPA) or alteplase was the first thrombolytic agent approved for the treatment of ischemic stroke. It is an enzyme that converts the plasminogen into plasmin, degrading the fibrin network that composes the thrombus. rtPA has proven efficacy in achieving successful recanalization, improving outcomes, and reducing disability, in approximately 30–50% of patients [3]. However, it has some drawbacks that limit its administration, such as a short half-life (~5 min), and it requires *bolus* and continuous infusion administration under clinical supervision. Furthermore, rtPA treatment is related to neurotoxic effects and the risk of hemorrhagic transformations and neuronal damage due to ischemia/reperfusion injury. All these issues limit its applicability, reducing the number of patients that can benefit from the treatment [4].

Nanotechnology has emerged as a promising tool to overcome rtPA drawbacks and increase its thrombolytic effect. The expectation regarding this technology in the stroke field is reflected in the literature with the design of a multitude of nanomedicines and approaches, which differ in composition, structure, and size, such as liposomes [5], polymeric nanoparticles [6,7,8], silica-coated magnetic nanoparticles [9], magnetic nanomedicines [10], magnetic microrods [11], gold nanospheres [12], microbubbles [13,14], and deoxyribonucleic acid (DNA) nanodevices [15]. 

Initially, synthesized nanoparticles, based on rtPA complexes camouflaging the molecule and liposomes, were mainly designed to increase the safety and efficacy of the rtPA-thrombolytic treatment [16,17,18]. Moreover, to increase efficacy by improving clot targeting, nanosystems have become increasingly sophisticated over the years. The use of antibodies against fibrin [19,20,21], peptides (L-arginine-glycine-aspartic acid peptide, RGD [22,23], L-arginine-glycine-aspartic acid-serine tetrapeptide, RGDS [13], or against factor XIII [24]), or other compounds (gelatin [25,26] or fucoidan [27,28]) that recognize specific components of the blood clot has made it possible to accumulate and focalize the treatment in the desired area [29]. In addition, controlled release of the drug has been achieved through the use of both internal stimuli, such as enzymes [30,31], pH [32], and temperature changes [33] or external stimuli, including ultrasound [25] or magnetic fields [34]. Thus, nanoparticles have enabled the reduction in the therapeutic dose of rtPA, extended its half-life, facilitated its administration, and reduced the risk of hemorrhages [35]. However, the use of this thrombolytic technology is currently limited by poor biocompatibility and stability, rapid clearance, or reduced targeting efficacy [36].

In recent years, the interest in cell membrane-derived (CMD) nanomedicines has increased due to their high biocompatibility, ability to avoid the phagocytic system, and specific clot targeting. These biomimetic nanoparticles have been fabricated from different cell types including neural stem cells, macrophages, red blood cells, neutrophils, and platelets [37]. Specifically, in the field of thrombolysis for ischemic stroke, the most common cells used have been platelets, as they have natural thrombus-targeting activity, interacting with collagen, von Willebrand factor, fibrin, or extracellular matrix. Moreover, platelets evade the clearance mediated by the immune system due to the CD47 receptor, which interacts with the signal regulatory protein (SIRPα), which when activated, downregulates phagocytosis [38]. Studies carried out in platelet membrane-derived nanomedicines for ischemic stroke have shown these nanoparticles to increase targeting efficiency and decrease the risk of bleeding as well as to reduce the infarct volume and allow treatment administration as a single *bolus* instead of continuous perfusion, such as is required for clinical rtPA administration [39,40,41,42,43,44,45]. Recently, a neutrophil mimetic nanosystem demonstrated high targeting efficacy and enhanced thrombolysis, anti-inflammation, and antioxidation in a preclinical model of ischemic stroke [46]. The successful results obtained with this type of nanosystem and their high biocompatibility would make this type of nanomedicine an excellent alternative to focus on in the future.

Despite the promising results of these nanomedicines and numerous articles published, there is currently not a general evidence overview in terms of meta-analyses evaluating whether thrombolytic nanoparticles are better than standard treatment with rtPA. Therefore, the main objective of this meta-analysis is to evaluate the efficacy of thrombolytic nanomaterials in ischemic stroke and to assess whether they offer advantages in terms of infarct volume and neurological scores compared to the conventional use of free rtPA in preclinical studies of ischemic stroke. It will also investigate whether there are differences between general nanoparticles and CMD nanomedicines. These analytical studies will support future research in this area and convince policymakers of the potential use of this nanotechnology for acute stroke treatment.

## 2. Materials and Methods

### 2.1. Search Strategy

A comprehensive literature search was performed according to the Preferred Reporting Items for Systematic Reviews and Meta-Analyses (PRISMA) 2020 Checklist (Appendix A) and the PRISMA 2020 Flow Diagram (Appendix A) [47].

We searched the Pubmed database for ((((stroke OR ischemic stroke OR ischaemic stroke) AND (drug delivery OR nanomedicine OR nanoparticles OR nanocapsules OR capsules OR nanomedicines OR nanosystems OR submicrometric OR sub-micrometric OR particles OR nanovesicles OR nanoformulation OR liposomes)) AND (thrombolytic therapy OR thrombus OR thrombolysis OR tissue plasminogen activator OR rtPA)) NOT (case report)) NOT (clinical trial). Titles and abstracts were initially screened by three authors (FL-G, NP-A, and CC-P).

No restrictions were placed on the language or date of publication, and the last search was conducted on 7 November 2024. The references of systematic reviews and eligible studies were revised, and suitable studies not detected previously were selected for inclusion assessment by the reviewers.

### 2.2. Inclusion Criteria, Data Extraction, and Risk of Bias

Full-text assessment of the selected papers and data extraction were conducted by two authors (FL-G and CC-P).

The selected publications in the first screening were checked to ensure they met the inclusion criteria. First, only studies investigating the efficacy of rtPA in nanoparticles or microparticles in preclinical models of exclusively cerebral ischemia were included. Due to the aim of this study being to determine the effectiveness of rtPA-containing nanosystems for ischemic stroke, when the thrombolytic treatment was not included in the nanostructure, those studies were discarded. Next, articles were included if lesion volume and/or neurological score were reported as outcome measures.

Information extracted included the species, stroke model, type of nanomaterial and whether the material was combined with another therapeutic agent, the time of administration, the dose of rtPA administered, the technique for measuring infarct volume, and functional test carried out.

To perform the analysis, mean, standard deviation (SD), and group sizes of lesion volume and neurological score were extracted. When studies represented the mean ± standard error of the mean (SEM), data were extracted to SD. Following previous studies [48], when data were only represented graphically, the online tool WebPlotDigitizer (https://automeris.io/WebPlotDigitizer/; accessed on 11 November 2024) was used to estimate mean and SD from graphs. To standardize the data, all infarct volumes were relativized to the control group (vehicle or surgery control).

Regarding neurological improvement, only publications that performed the neurological scale as a behavioral test were included in the analysis. When multiple outcome measures were included at different time points, the end point was selected.

According to the clinical practice, intravenous administration of free rtPA was chosen as the control group for comparison, as the thrombolytic effect of nanosystem administration was the focus of this analysis. A study was defined as a specific comparison between treatment (with nanosystems) and control group (treated with rtPA). In case a publication presented more than one specific result (e.g., infarct lesion volume and neurological scoring), each one of them was included in the meta-analysis as a single study. Moreover, some publications investigated different administration times, and each of them was included as a different study.

The Preclinical Evidence of Readiness in Stroke Models Evaluating Drugs and Devices (PRIMED^2^) assessment tool evaluates studies for the likelihood of success in future clinical trials [49]. This has been carried out in order to evaluate the articles according to the degree of translatability of the selected publications. It takes into account 11 items that are scored from 0–2 (Appendix A): (1) sex of animals; (2) age of animals; (3) species and strains of animals; (4) reproducibility; (5) treatment time epoch; (6) baseline comorbidities; (7) feasible time window; (8) dose-response; (9) feasible route of delivery; (10) behavioral and/or long-term outcome assessment; (11) typical infarct volume reduction magnitude. PRIMED^2^ scores range from 0 to 22 points, with 0–7 being low, 8–15 intermediate, and 16–22 high translatability.

The risk of bias of each publication was assessed using CAMARADES (Collaborative Approach to Meta-analysis and Review of Animal Data in Experimental Studies) study quality checklist [50]: (1) peer-reviewed publication; (2) control of temperature; (3) random allocation to treatment or control; (4) blinded induction of ischemia; (5) blinded assessment of outcome; (6) use of anesthetic without significant intrinsic neuroprotective activity; (7) animal model (aged, diabetic, or hypertensive); (8) sample size calculation; (9) compliance with animal welfare regulations; and (10) statement of potential conflict of interests. Studies were divided into low risk of bias when their CAMARADES score was greater than or equal to 5 and high risk of bias when the score was less than 5.

Concerning subgroup analysis, results of infarct volume were divided into type of nanomaterial, administration time, type of stroke model, and risk of bias. Regarding the type of nanosystem, CMD biomimetic nanomedicines were separated from the rest of the nanoparticles. In the “nanoparticles” group, polymeric nanoparticles, DNA devices, gold nanostars, gold nanospheres, magnetic microrods, microbubbles, and discoidal particles were included. The CMD nanomedicines group included neutrophil and platelet-derived particles. Furthermore, according to the administration time, publications were divided into early administration (from 10 to 30 min after ischemia) and late administration (from 1 to 6 h after ischemia). In publications with multiple time points of administration, data from each group were extracted separately. Concerning the stroke models, studies were divided into platelet-rich thrombus, fibrin-rich thrombus, and intraluminal filament models. Finally, selected articles were divided into high and low risk of bias depending on their CAMARADES score.

### 2.3. Statistical Analysis

First, we computed the standardized mean difference (SMD) of each individual study using Hedges’ g unbiased estimator, as it performs better for small sample sizes [51], and then computed pooled effect sizes, which were 95% weighted by the inverse of their variance [52]. We computed pooled effect sizes both by infarct lesion volume and neurological scoring.

SMDs are used as summary statistics when all the studies assess the same outcome but use different estimates or measures, such as different neurological scores. SMDs compare rtPA and nanosystems and are weighted by the pooled SD of both arms. An SMD of zero means no difference between both arms, a negative result favors nanoparticles in our case (as the infarct area decreases or neurological score improves versus the rtPA treatment), and a positive result favors rtPA. A large effect is defined as >0.8 units, a medium effect as >0.5 units, and a small effect as <0.2 units [53].

We assessed the presence of heterogeneity using the Dersimonian and Laird Q-test, and then quantified calculating the proportion of the total variance due to between-study variance (I^2^). We computed both fixed effects and random effects for pooled estimates, but as heterogeneity was present, we only reported the latter.

In addition to the main analysis, we performed several *a priori* planned subgroup analyses by nanosystem type, by administration time, and by risk of bias.

Publication bias was first assessed visually using funnel plots and then more formally using Egger’s regression test [54]. We also tested the potential effect of any publication bias by using the trim-and-fill method described by Duval and Tweedie [55]. All analyses were performed using Stata version 17 (StataCorp LP, College Station, TX, USA).

## 3. Results

### 3.1. Database Search

Figure 1 shows the results of the search strategy. We identified 732 publications, of which 2 were excluded because they were retracted. After an initial evaluation, we excluded 703 publications because they did not deal with stroke, nanoparticles, or rtPA, they were in vitro studies only, or rtPA was administered as an adjuvant. Of 27 fully revised articles, 11 were discarded because they did not investigate thrombolysis in models of cerebral ischemia or rtPA was administered as an adjuvant. After reviewing the references of the articles that met the inclusion criteria and systematic reviews, two additional articles were included [11,14]. Finally, 18 papers were included in the meta-analysis.

### 3.2. Study Characteristics

All publications that met the inclusion criteria were published between 2018 and 2024 (Figure 2A). The number of publications related to CMD nanomedicines has increased, particularly in recent years. The characteristics of the publications are shown in Appendix A, including animal species, number of animals used, stroke model, biomaterial and size, whether the nanosystems were combined with other drugs or external stimuli, rtPA dose, treatment time points, route of administration, and how and when lesion volume and functional assessments were performed.

A total of 200 animals (treated with free rtPA, simple size *n* = 100; treated with nanosystems, simple size *n* = 100) were included in the meta-analysis reporting 28 comparisons. Of these, 20 assessed lesion volume and 8 neurological score.

All publications included were conducted with rats (*n* = 3) or mice (*n* = 15) (Figure 2B). The most common ischemic stroke models used were the phototrombotic distal occlusion model employing rose Bengal or rose Bengal with thrombin (*n* = 7), followed by the intraluminal filament middle cerebral artery occlusion (MCAo) (*n* = 5), thromboembolic model using thrombin (*n* = 4), and FeCl_3_ (*n* = 2) (Figure 2C). The techniques used to measure the infarct volumes were magnetic resonance imaging (MRI), 2,3,5-triphenyltetrazolium chloride (TTC), laser speckle imaging (LSI), and optical coherence tomography (OCT) (Figure 2D). During the analysis, the study performed by Xu J et al. [44] was excluded from the infarct volume analysis, as no quantification of the lesion was carried out. Furthermore, the experiment with 6 h of administration after ischemia in the research of Yin J et al. [15] was excluded because there was no control group.

The assessment of neurological deficit was performed using different tests (Figure 2E,F): modified neurological severity score (4, 5, 14, and 18 points) (*n* = 8), CatWalk CT (*n* = 3), RotaRod, Morris water maze, beam walking, and cylinder tests. Eight publications did not perform any functional test, six articles conducted one test, and only four carried out more than one functional assessment. In this meta-analysis, only neurological scores were used to perform the analysis, and the studies carried out by Mei T et al. [7] and Choi W et al. [10] were ruled out from the neurological outcome analysis because they used only the cylinder and RotaRod, respectively.

The PRIMED^2^ was designed to score articles based on how translatable the preclinical study might be to the clinic. The PRIMED^2^ score in this meta-analysis showed that 10 of 18 articles had a low translation readiness and 8 corresponded to intermediate readiness (Appendix A).

### 3.3. Analysis of Efficacy

Meta-analysis was performed on lesion volume and neurological assessment data. The heterogeneity of the studies was high for both infarct volume (I^2^ = 82.94%) and neurological scales (I^2^ = 66.66%).

A high variability in the effect of the nanosystems on lesion volume was observed (ranging from 1.18 to −10.58), but the results showed a reduction in the infarct volume compared to the standard treatment with rtPA (SMD: −2.59, 95% CI: −3.49, −1.69) (Figure 3).

Importantly, two of the included trials [26,43] showed positive SMD, suggesting a better outcome in the rtPA-treated groups. In the case of the study by Correa-Paz C et al., the nanoparticles were found to be ineffective, causing an increase in infarct volume (SMD: −1.18, 95% CI: −0.05, 2.42). The research by Miggliavaca M et al. reported a similar reduction in infarct volume in the groups treated with rtPA and CMD nanomedicines, with a slightly positive SMD (SMD: 0.23, 95% CI: −0.70, −1.16).

Figure 4 shows that treatment with nanosystems improved neurological outcomes (SMD: −2.92, 95% CI: −3.95, −1.89), presenting a lower variability than in the infarct volume results (ranging from −1.35 to 5.77).

### 3.4. Cell Membrane-Derived Nanomedicines Improve Infarct Volume Compared to Nanoparticles

In the first subgroup analysis, the publications were divided according to the type of nanoparticle, which are summarized and organized by their size in Figure 5. The group called nanoparticles includes polymeric nanoparticles, DNA devices, gold nanostars, gold nanospheres, magnetic microrods, microbubbles, and discoidal particles. Three studies [10,11,14] used micrometer-sized particles. CMD nanomedicines are synthesized from platelet and neutrophil membranes.

Table 1 summarizes the effect of the two subtypes of nanosystems on lesion volume. CMD nanomedicines showed a greater decrease in infarct volume than the nanoparticles, compared with the group treated with rtPA.

### 3.5. Nanosystems Extend the Therapeutic Window

To analyze the effect of the time of administration from stroke onset, the studies were divided into early (0–30 min after the onset of ischemia) and late administration (30 min–3 h after the onset of ischemia). Most studies with late delivery used CMD nanomedicines (Figure 6).

Table 2 represents results on infarct volume depending on the time of administration. Late delivery was associated with a greater reduction in infarct volume.

### 3.6. Stroke Models

The studies were divided according to the infarct model used, with three groups using platelet-rich thrombi (photothrombotic and FeCl_3_), fibrin-rich thrombi (thrombin model), and intraluminal filament models.

Table 3 summarizes the effect of the stroke model, showing that the filament model produced the greatest reductions in infarct volume, followed by the platelet-rich and thrombin-rich models.

### 3.7. Risk of Bias

The articles were classified as low or high risk according to the score obtained in the CAMARADES list (Appendix A). The high-risk bias group shows a higher recovery of animals compared to the low-risk group (Table 4).

### 3.8. Publication Bias Analysis

The funnel plot of the studies included in the meta-analysis (Figure 7) showed slightly asymmetric toward the left, that is, toward the nanoparticle effect side, which was confirmed by the quantitative Egger test (*p* = 0.0001). Notwithstanding, the trim-and-fill test, after imputing six studies yielded a statistically significant corrected effect size of −1.56 (−2.49, −0.64).

## 4. Discussion

To our best knowledge, this is the first meta-analysis of studies analyzing the therapeutic effect and the neurological outcomes of nanoparticles encapsulating rtPA in preclinical models of ischemic stroke. A total of 18 publications met our inclusion criteria, suggesting that nanoparticles designed to achieve thrombolysis in stroke improved infarct volume and neurological scores.

The results of this meta-analysis suggest that both nanoparticles and CMD nanomedicines confer an advantage over standard thrombolytic therapy with rtPA. This is consistent with published articles reporting prolonged drug half-life [6,7,43], reduced bleeding risk [7,39,40,43], faster administration [42,43], window of opportunity [10], and improved targeting of the clot [15,39,40,41,42].

A particular benefit of CMD nanomedicines in reducing infarct volume over other kinds of nanoparticles was observed. This can be attributed to both their biological and physicochemical properties. In addition to their excellent biocompatibility and targeting capabilities, the size of these nanosystems, ranging from 140 to 220 nm, is related to longer circulation times. Small nanoparticles can be filtered by the kidneys, whereas larger nanoparticles can be trapped in capillaries. We also found that CMD nanomedicines have a neutral or negative zeta potential, which is also associated with longer circulation times, as the adsorption of proteins that facilitate removal by the reticuloendothelial system is reduced [56]. Larger bioavailability can be responsible, at least in part, for greater efficacy.

Late administration of rtPA is associated with lower recanalization rates, resulting in higher infarct volumes. In our meta-analysis, we observed a higher benefit of nanomedicines that were administered at late times. It is important to note that most of the studies in which nanoparticles were administered after the first half hour were conducted in CMD nanomedicines (Figure 6), which have a higher bioavailability and may allow for longer therapeutic windows.

The publications analyzed showed variability in the stroke models used, the quantification of infarct volume, the neurological scales used, and the doses of rtPA used, which may influence the heterogeneity of the meta-analysis results. On one hand, it is important to use an appropriate preclinical stroke model when studying thrombolytic therapy. Most of the studies reviewed used the photothrombotic or FeCl_3_ model, which produces a platelet-rich thrombus that does not respond to rtPA [57], or the intraluminal filament MCAo model, which is used to study neuroprotective drugs rather than pharmaceutical thrombolysis [58]. Currently, the fibrin-rich thromboembolic stroke model is widely accepted for testing new thrombolytic approaches. This model, induced by an in situ injection of thrombin in the middle cerebral artery (MCA), appears to be pathophysiologically closer to a human stroke and is widely accepted to evaluate the recanalization effects of rtPA [59,60]. Curiously, in this meta-analysis, we found that the models associated with better evolution were the MCAo model and platelet-rich thrombus models (rosa Bengal and FeCl_3_). This could be due to the fact that, when stratified by infarct model, we significantly reduced the sample to be analyzed, obtaining, for example, *n* = 4 in the thrombin model. It is important to note that one of these studies has a negative result [26] and the other has an effect similar to that of standard thrombolytic therapy with CMDs [43]. Another parameter to take into account is the previous experience of the research group in developing stroke models, which introduces variability and bias into the data.

In our study, lesion volume was chosen as the outcome measure, as this is the main focus of preclinical studies in ischemic stroke. It should be noted that the time points and techniques used to measure infarct volume vary between studies. For example, although most measurements are made at 24 h, in some cases, this is delayed to 3 or 7 days. The techniques used to measure infarct volumes included histological techniques such as TTC (8 trials) and MRI (9 trials). The choice of technique depends on the availability of scanners that provide more detailed images. However, all this adds to the variability between studies, which we have tried to overcome by relativizing the infarct volumes to the control group.

In the reviewed studies, we found variability in the test used to assess neurological recovery and in the time at which the tests were carried out. In the case of the neurological scales, our analysis selected the last measurements taken in each study, ranging from 24 h to 8, 14, or 20 days. Another limitation of the motor scores is the high recovery of the rodents after cerebral ischemia, which does not allow the detection of deficits. Furthermore, motor deficits should not be observed in models involving only the cerebral cortex of rats and mice, which are the majority of models used in the selected studies. This is because, in contrast to humans, the rodent cerebral cortex only minimally involves the primary motor cortex and, instead, covers the barrel sensory, auditory, and visual sensory areas [61].

Regarding biases, the CAMARADES checklist was used to evaluate the risk of bias depending on the number of items fulfilled by each study. It is important to remark that no single article included all CAMARADES items throughout the study. In fact, the aged, diabetic, or hypertensive animal models were not used, and the sample size calculation was not performed in any case. Moreover, only one paper blindly induced ischemia [42]. When the CAMARADES score is less than 5, there is a high risk of bias, while if it is equal to or greater than 5, the risk of bias is low. In our study, we found that 6 of 17 investigations had a low risk of bias, which is a greater improvement in infarct volume than the high-risk studies. This is in accordance with a meta-analysis performed by O’Collins et al. [62] about neuroprotectants, which demonstrated that, in preclinical studies of ischemic stroke, trials with better therapeutic outcomes had lower study quality scores, attributable to bias due to a lack of randomization and blinding. Furthermore, this may explain the lack of transition from bench to bedside and highlights the need to be more rigorous and cautious with the results obtained. However, although we have observed that there is a positive trend in terms of the therapeutic effect and neurological improvement, there is a risk of bias that makes us careful with the results.

Furthermore, we identified a potential publication bias in our results based on the funnel plot analysis, which was confirmed by the results of the Egger test. This kind of selection bias occurs when studies with statistically significant or favorable results are more likely to be published at the expense of those with negative or null results, leading to an overestimation of the true effect size in meta-analyses. To address this question, we made extensive efforts to minimize this bias by performing a comprehensive search strategy. Specifically, we did not restrict our search by country or language, and we employed broad search terms to ensure a wide capture of relevant studies, as commented previously. Additionally, we performed the trim-and-fill analysis, a method that estimates and imputes missing studies to adjust for potential publication bias by symmetrizing the funnel plot. Although the analysis imputed six studies, the adjusted effect size remained statistically significant, reinforcing the robustness of our findings. Although it is known that publication bias is one of the major problems in the reporting of basic research, Sena et al. demonstrated that one in seven experiments are not published; this results in an exaggeration of the efficacy of treatments [63]. Thus, animals that have been used in studies with negative results are not adding to human knowledge. On the other hand, participants in clinical studies may be affected by this overestimation of results, which is also an ethical problem [64]. The ease of publishing negative and neutral results should be a priority for the scientific community.

It can be observed that the included studies have been published in recent years, since 2018, with the highest number of publications in the last three years. This reflects the interest of the scientific community and their relevance for the nanothrombolytic therapy in ischemic stroke. Despite promising results, none of these preclinical studies have progressed to the next preclinical stages or been transferred to patients [65,66]. There are several reasons for this, including the challenges of studying nanomaterials, the experimental design of preclinical studies that can hinder the transition from the laboratory to the clinic, and clinical aspects such as the approval and marketing of new drugs for ischemic stroke.

When encapsulating drugs such as rtPA, it is not only important to use methods that do not affect the stability and activity of the drug but also to ensure that the physico-chemical properties of the nanoparticles remain stable during scale-up, which is a critical aspect of their future pharmaceutical manufacturing [67]. Another key process is the preservation of the nanoparticles, especially in the case of rtPA, a molecule that is particularly sensitive to temperature changes and has reduced solubility. Techniques such as lyophilization are frequently used to increase the shelf-life of nanomedicine formulations, ensuring the long-term preservation of nanoparticles and facilitating their handling and storage [43]. In addition, the toxicity of nanosystems on different cell types should be thoroughly investigated. Biodistribution and pharmacokinetics should also be studied to help reduce the side effects associated with these drugs. To enable progress toward clinical trials, not only should more toxicity, biodistribution, and pharmacokinetic studies be carried out, but also further research into scaling and preservation of nanosystems is increasingly needed.

For preclinical research in ischemic stroke, guidelines such as Animal Research: Reporting of in vivo Experiments (ARRIVE) [68] or the Stroke Therapy Academic Industry Roundtable (STAIR) recommendation [69] guidelines allow the establishment of rigorous and detailed standards that increase the reproducibility and quality of the studies. This reduces bias and increases the validity of the results obtained, making the studies comparable and increasing the likelihood of successful translation to patients. These guidelines include a series of recommendations to be included in the studies, such as a description of the animals and experimental procedures, a justification of the sample size, randomization and blinding, and comorbidities (including aging), among others. The PRIMED^2^ assessment has been developed as a way of assessing certain aspects of the above guidelines. It scores studies according to their likelihood of translation to the clinic, evaluating items such as the number of species, sex, and age of the animals, multicentric studies, or feasible routes of delivery [49]. The majority of the studies included in this meta-analysis scored low on the PRIMED^2^ checklist, highlighting the need for more careful design of preclinical studies.

From a clinical point of view, the translation of nanosystems in the ischemic stroke field differs from other areas such as oncology or infectious diseases, where many formulations such as liposomes and polymeric nanoparticles have been commercialized since 1995 [70]. In the field of neurology, there are a few clinical trials, in particular, with liposomes for the treatment of brain tumors or degenerative diseases, but they have not yet been approved by the regulatory authorities [71]. It is important to note that if we look at the development of rtPA treatment, it was approved in 1995 for ischemic stroke but has been commercialized since 1987 for the treatment of myocardial infarction. The same is true for another thrombolytic drug, tenecteplase, which has been approved for cardiology since 2000 and has shown good results in clinical stroke trials but has not yet been approved [72]. Due to the high variability of stroke patients and the narrow therapeutic window in the acute phase, clinical trials to validate new drugs are challenging. It should also be noted that the brain is a very complex and vulnerable organ where the effects of minimally toxic compounds can be devastating. In particular, nanomedicine is still under-regulated by the Food and Drug Administration (FDA) and the European Medicines Agency (EMA), and the regulatory processes are time-consuming and require a high level of expertise. Increased regulation in this area will help to refine and standardize the requirements for the approval of safe nanomedicine products [73].

### Limitations

There are some limitations to our study. Firstly, the number of articles included is relatively small. This is due to two reasons: the search and inclusion criteria. When we searched Pubmed for articles synthesizing nanoparticles with rtPA as a treatment for ischemic stroke, we found that the terminology used to refer to nanosystems varied widely. For example, terms such as nanoparticle, nanocapsule, nanomedicine, nanosystem, and nanovesicle, among others, can be found. This is why the terms in our search are so extensive. The lack of standardization to refer to different types of nanoparticles makes it difficult to find relevant information. Keywords should allow researchers to efficiently find studies of interest, facilitate comparison between studies, and avoid confusion caused by terminological diversity. This may have meant that we missed some articles that would have been included in the meta-analysis.

About the inclusion criteria, despite the multitude of nanoparticles that have been developed in recent years, few have been tested in preclinical studies. More specifically, within the preclinical studies, we have only selected nanoparticles whose therapeutic effect has been proven in ischemic stroke, thus discarding some studies in which the therapeutic effect was tested on thrombi in the iliac [21], mesenteric [74], carotid [75], femoral [13], or abdominal aorta [76] arteries.

On the other hand, some data were estimated with the graphs, which could result in a difference in the collected data versus the real data. However, following previous meta-analysis, the differences between raw and estimated data are minimal [48,77], which should not influence the conclusions of our work.

Finally, we found the presence of high between-studies heterogeneity in our analyses. Some of the reasons that may explain this feature are related to the compilation of studies that are different in design, terms of rtPA doses, administration times, and types of nanosystems used, which is expected to a certain point. To overcome these differences, subgroup analyses have been carried out in terms of administration times, differentiating between early administration (less than or equal to 30 min) and late administration (greater than or equal to one hour). In addition, we have differentiated CMD nanomedicines from the polymeric and inorganic nanoparticles and carried out a subgroup analysis by the risk of bias. Furthermore, we based the interpretation of our results on the random-effect estimates, as recommended.

## 5. Conclusions

Despite the cited limitations and the high variability in the included studies, this meta-analysis suggested that nanomedicine is promising for improving thrombolytic therapy for ischemic stroke, showing a reduction in infarct volume and better neurological improvement in preclinical studies using different types of nanoparticles.

However, the biases we have encountered highlight the need to be more rigorous in following guidelines for recommendations in animal practice as well as the need for all positive, negative, or neutral results to be published. Even so, the encapsulation of thrombolytic therapy, especially CMD nanomedicines, offers an advantage over standard treatment, which should be further investigated using biocompatible nanomedicines.

## Figures and Tables

**Figure 1 pharmaceutics-17-00208-f001:**
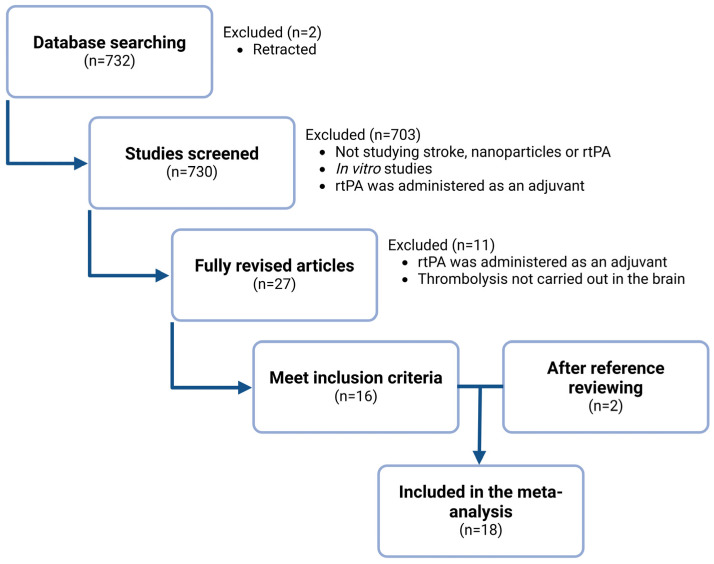
Diagram summarizing the literature search and the studies excluded and included in the meta-analysis.

**Figure 2 pharmaceutics-17-00208-f002:**
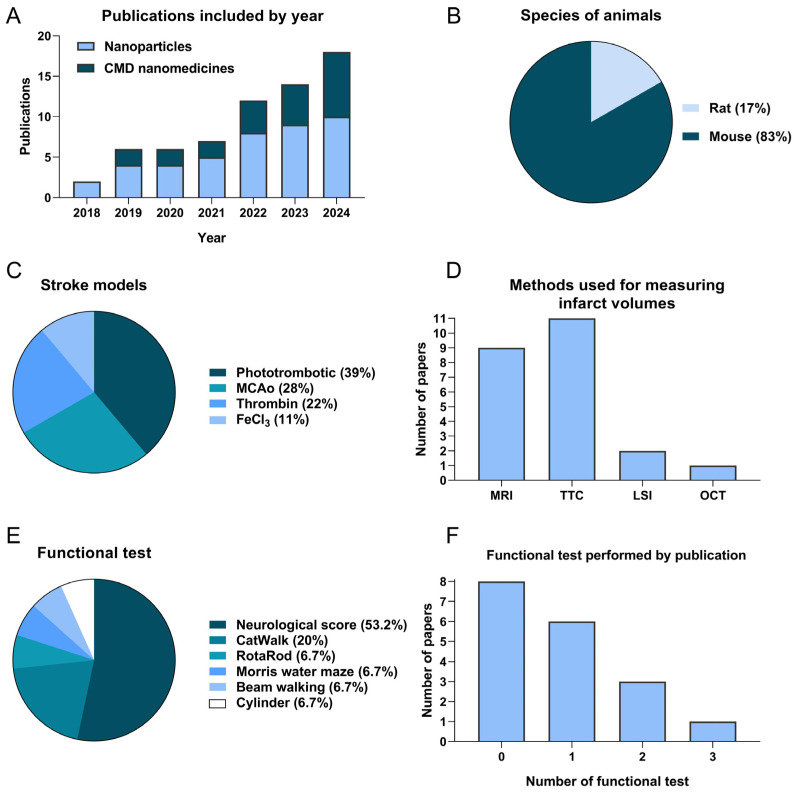
Description of characteristics of the studies included in the meta-analysis. (**A**) Accumulative studies included in the study by year of publication. (**B**) Number of papers using each species. (**C**) Type of ischemia models and (**D**) methods used to measure the infarct volume in the included studies. (**E**) Type of functional test and (**F**) number of functional tests performed by study. LSI, laser speckle imaging; MRI, magnetic resonance imaging; OCT, optical coherence tomography; TTC, 2,3,5-triphenyltetrazolium chloride.

**Figure 3 pharmaceutics-17-00208-f003:**
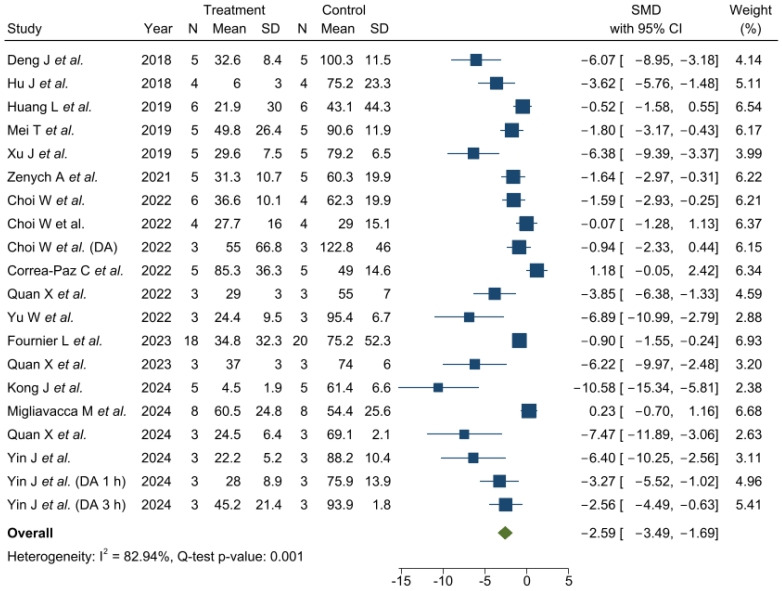
Effect size of nanosystems for lesion volume. Forest plot of standardized mean difference and 95% CI. CI, confidence interval; SMD, standardized mean difference. Deng J et al. [6], Hu J et al. [11], Huang L et al. [8], Mei T et al. [7], Xu J et al. [45], Zenych A et al. [28], Choi W et al. [12], Choi W et al. [10], Correa-Paz C et al. [26], Quan X et al. [39], Yu et al. [40], Fournier L et al. [14], Quan X et al. [41], Kong J et al. [42], Migliavacca M et al. [43], Quan X et al. [46], Yin J et al. [15].

**Figure 4 pharmaceutics-17-00208-f004:**
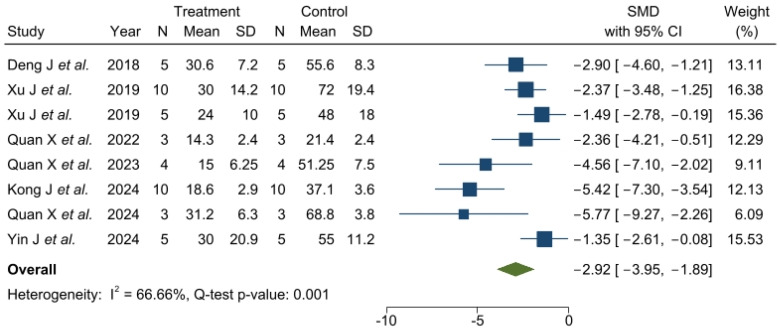
Effect size of nanosystems for neurological scoring. Forest plot of standardized mean difference and 95% CI. CI, confidence interval; SMD, standardized mean difference. Deng J et al. [6], Xu J et al. [44], Xu J et al. [45], Quan X et al. [39], Quan X et al. [41], Kong J et al. [42], Quan X et al. [46], Yin J et al. [15].

**Figure 5 pharmaceutics-17-00208-f005:**
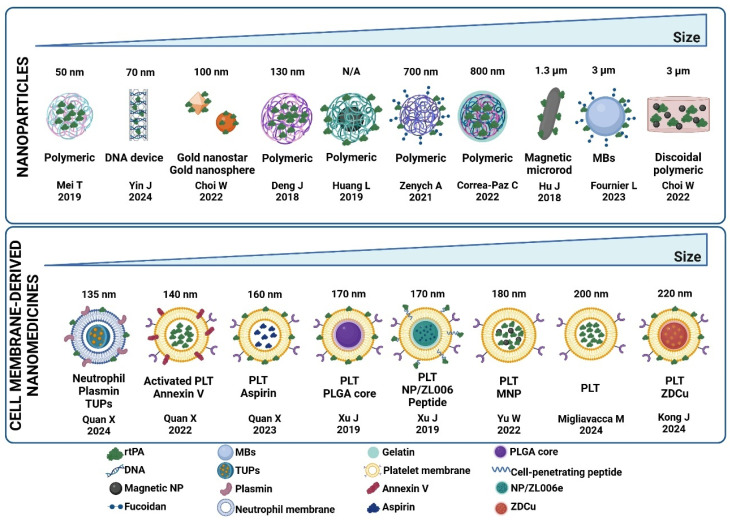
Nanoparticles and cell membrane-derived nanomedicines included in the meta-analysis. DNA, deoxyribonucleic acid; MBs, microbubbles; NP, nanoparticle; PLGA, poly(lactic-co-glycolic) acid; PLT, platelet; rtPA, recombinant tissue plasminogen activator; TUPs, thylakoid membrane-coated upconversion nanoparticles; ZDCu, zein + docosahexaenoic acid + selenium nanoparticle. BioRender (https://biorender.com/) was used for creating the figures (https://biorender.com/; access date: 20 November 2025). Mei T et al. [7], Yin J et al. [15], Choi W et al. [12], Deng J et al. [6], Huang L et al. [8], Zenych A et al. [28], Correa-Paz C et al. [26], Hu J et al. [11], Fournier L et al. [14], Choi W et al. [10], Quan X et al. [46], Quan X et al. [39], Quan X et al. [41], Xu J et al. [44], Xu J et al. [45], Yu et al. [40], Migliavacca M et al. [43], Kong J et al. [42].

**Figure 6 pharmaceutics-17-00208-f006:**
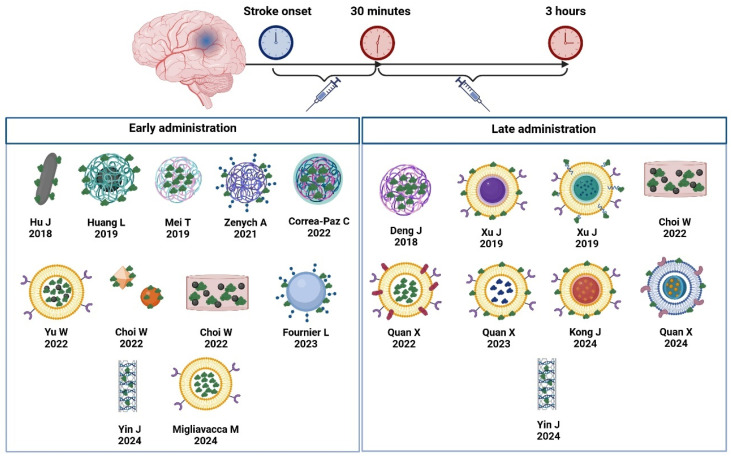
Nanoparticles and cell membrane-derived nanomedicines organized depending on their early or late administration. BioRender (https://biorender.com/; access date: 20 November 2024) was used for creating the figures. Hu J et al. [11], Huang L et al. [8], Mei T et al. [7], Zenych A et al. [28], Correa-Paz C et al. [26], Yu et al. [40], Choi W et al. [12], Choi W et al. [10], Fournier L et al. [14], Yin J et al. [15], Migliavacca M et al. [43], Deng J et al. [6], Xu J et al. [44], Xu J et al. [45], Quan X et al. [39], Quan X et al. [41], Kong J et al. [42], Quan X et al. [46].

**Figure 7 pharmaceutics-17-00208-f007:**
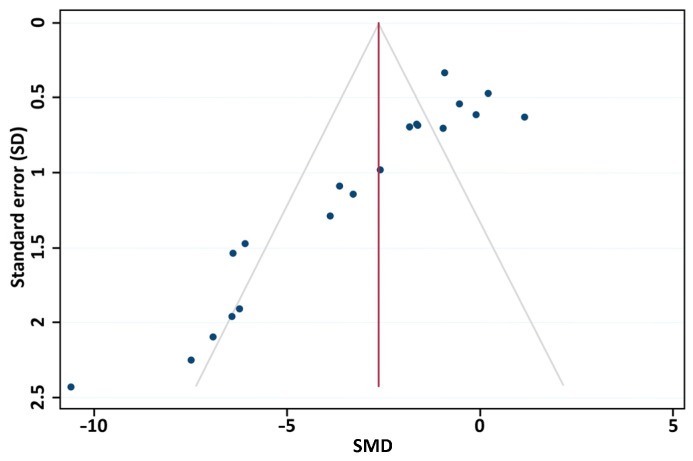
Funnel plot of observed studies. The blue dots represent the SMD of each individual study, and the red line the overall effect size. SMD, standardized mean differences.

**Table 1 pharmaceutics-17-00208-t001:** Subgroup comparing type of nanosystem on lesion volume. CI, confidence interval; CMD, cell membrane-derived; SMD, standardized mean differences.

Lesion Volume			
Type of Nanosystem	Studies	SMD (95%, CI)	I^2^, Q-Test *p*-Value
Nanoparticles	13	−1.65 (−2.47, −0.84)	75.75, *p* = 0.0001
CMD nanomedicines	7	−5.61 (−8.97, −2.25)	90.04, *p* = 0.0001

**Table 2 pharmaceutics-17-00208-t002:** Subgroup comparing administration time on lesion volume. CI, confidence interval; SMD, standardized mean differences.

Lesion Volume			
Administration Time	Studies	SMD (95%, CI)	I^2^, Q-Test *p*-Value
Early administration (0–30 min)	11	−1.24 (−2.09, −0.4)	77.57, *p* = 0.0001
Late administration (30 min–3 h)	9	−4.74 (−6.55, −2.93)	76.57, *p* = 0.0001

**Table 3 pharmaceutics-17-00208-t003:** Subgroup comparing stroke models. CI, confidence interval; CMD, cell membrane-derived; SMD, standardized mean differences.

Lesion Volume			
Stroke Model	Studies	SMD (95%, CI)	I^2^, Q-Test *p*-Value
Platelet-rich thrombus	10	−2.43 (−3.58, −1.27)	75.23, *p* = 0.001
Fibrin-rich thrombus	4	−0.29 (−1.35, 0.77)	78.02, *p* = 0.001
Intraluminal filament	6	−5.34 (−7.38, −3.31)	66.29, *p* = 0.01

**Table 4 pharmaceutics-17-00208-t004:** Subgroup comparing risk of bias on lesion volume. CI, confidence interval; SMD, standardized mean differences.

Lesion Volume			
Risk of Bias	Studies	SMD (95%, CI)	I^2^, Q-Test *p*-Value
High risk	11	−3.22 (−4.50, −1.95)	82.25, *p* = 0.0001
Low risk	6	−1.97 (−3.81, −0.13)	87.35, *p* = 0.0001

## Data Availability

Data sharing is applicable to this article.

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
