# Peer review of "Nanoparticles for Thrombolytic Therapy in Ischemic Stroke: A Systematic Review and Meta-Analysis of Preclinical Studies"

_pharmaceutics, 2025, doi:10.3390/pharmaceutics17020208_

Round 1
Reviewer 1 Report
Comments and Suggestions for Authors
The authors of the manuscript performed a systemic review of the preclinical studies using nanoparticles as a carrier for thrombolytic therapy (the recombinant tissue plasminogen activator (rtpA) or alteplase) in ischemic stroke treatment. Analytic results from rodent experiments of 18 publications concluded that the approach is promising for improvement of ischemic stroke treatment using rtpA, by showing a reduction in infarct volume and improved neurological outcomes in preclinical studies. Among the different types of nanoparticles, cell membrane-derived (CMD) nanomedicines seem to provide the best beneficial effect. Overall, the manuscript was well organized, and the conclusion is straightforward. However, the reviewer would like to recommend the following revision before its publication:
(1) Although both rats and mice are rodents, many in vivo data from the two species are not the same. Including data from both rats and mice to do meta-analysis should be rationalized.
(2) What is half-life of the CMD nanomedicine (i.e. platelet membrane encapsulated rtPA) in comparison to the half-life of rtpA in standard treatment.
(3) Except for simply presenting data, the authors should comment on the reasons why the results from refs [26, 43] in Figure 3 showed larger lesion volumes in comparison to these in the control groups.
(4) As a systematic review article, the authors are responsible for reporting and analyzing all relevant data published by far. It is hard to agree with a statement that due to difficulty finding relevant information some articles that should have been included would be missed in the meta-analysis. In principle, if the number of missed articles is large and all provided opposite results. The conclusion from the current manuscript does not hold and is meaningless.
Author Response
Decision on the Manuscript ID pharmaceutics-3431099 for Pharmaceutics
Thank you for your consideration of our article in Pharmaceutics journal and for the positive recommendations made by the Reviewers prior to acceptance for publication.
Changes in the MS and SI have been labeled in yellow.
Authors’ response to Reviewer 1 comments
The authors of the manuscript performed a systemic review of the preclinical studies using nanoparticles as a carrier for thrombolytic therapy (the recombinant tissue plasminogen activator (rtpA) or alteplase) in ischemic stroke treatment. Analytic results from rodent experiments of 18 publications concluded that the approach is promising for improvement of ischemic stroke treatment using rtpA, by showing a reduction in infarct volume and improved neurological outcomes in preclinical studies. Among the different types of nanoparticles, cell membrane-derived (CMD) nanomedicines seem to provide the best beneficial effect. Overall, the manuscript was well organized, and the conclusion is straightforward. However, the reviewer would like to recommend the following revision before its publication:
We thank the Reviewer for her/his positive and kind comments.
(1) Although both rats and mice are rodents, many in vivo data from the two species are not the same. Including data from both rats and mice to do meta-analysis should be rationalized.
We agree with the reviewer’s comment. We have not previously performed this stratification according to other preclinical meta-analyses (Front Neurol. 2019 Aug 27:10:924; . The main objective of this meta-analysis is the efficacy of nanoparticles in preclinical models of ischaemic stroke, so although they are not the same species, the pathophysiological mechanisms are similar and the same types of stroke models are used in the studies.
We have performed a new subgroup analysis of possible differences between rats and mice. The results are as follows:
|
Lesion volume |
|||
|
Animal specie |
Studies |
SMD (95%, CI) |
I2, Q-test p-value |
|
Rat |
5 |
-5.18 (-7.52, -2.84) |
80.37, p=0.00001 |
|
Mice |
15 |
-1.84 (-2.71, -0.97) |
69.62, p=0.01 |
The subgroup of studies using rats shows a greater improvement in infarct volume, but it is important to note the difference in the number of studies. Having such a small sample compared to the other group may lead to selection bias, as small samples are associated with more extreme results.
(2) What is half-life of the CMD nanomedicine (i.e. platelet membrane encapsulated rtPA) in comparison to the half-life of rtpA in standard treatment.
The half-life of rtPA is approximately 5 minutes (analysed and reported in our previous studies, see for example: J Cereb Blood Flow Metab. 2024 Aug;44(8):1306-1318). It is important to note that few of the included studies performed a pharmacokinetic study that allows in-depth analysis of these data. In the case of the CMD nanomedicines reviewed for this meta-analysis, only 4 out of 8 studies include a pharmacokinetic study. In the published results, the half-life varies between 10 (Adv Mater. 2020 Jan;32(4):e1905145) and 2 (J Nanobiotechnology. 2024 Jan 3;22(1):10) times longer than free rtPA.
(3) Except for simply presenting data, the authors should comment on the reasons why the results from refs [26, 43] in Figure 3 showed larger lesion volumes in comparison to these in the control groups.
As the Reviewer remarks, in these two studies the lesion volumes were higher than in the rtPA control group. Specifically, these studies were carried out by our investigation groups. In the case of the reference 26 (J Nanobiotechnology. 2022 Jan 21;20(1):46), nanocapsules were found to be retained in the ischemic region, increasing infarct volume. This demonstrates that the size and composition of nanomedicines is a crucial factor in improving thrombolytic therapy. About the reference 43 (J Nanobiotechnology. 2024 Jan 3;22(1):10), there were almost no differences between the group treated with CMD nanomedicines and rtPA, although there were significant differences compared to the control group. However, this work showed that a very similar therapeutic effect was achieved with a simple bolus administration instead of perfusion.
Following the Reviewer’s suggestion we have included an explanation in the results (line 290-295).
(4) As a systematic review article, the authors are responsible for reporting and analyzing all relevant data published by far. It is hard to agree with a statement that due to difficulty finding relevant information some articles that should have been included would be missed in the meta-analysis. In principle, if the number of missed articles is large and all provided opposite results. The conclusion from the current manuscript does not hold and is meaningless.
We thank the Reviewer for her/his kind comment. Every systematic review, and therefore every meta-analysis, may be susceptible to publication bias, that is, not including all eligible studies, so analyses have been carried out to determine and evaluate the possible impact of this type of bias. In addition, we have detected an erratum in the manuscript that should be pointed out. In the paragraph referring to publication bias, lines (352-356) of the manuscript, it states “Notwithstanding, the trim-and-fill test, after imputing 26 studies (...)”. In reality, the trim and fill study imputed 6 studies, so there were 26 “observed + imputed” studies in total included in the analysis. We corrected the paragraph and added an explanation in the discussion (line 442-453).

Reviewer 2 Report
Comments and Suggestions for Authors
This study is of great importance as ischemic stroke remains one of the leading causes of death and disability worldwide. Recombinant tissue plasminogen activator (rtPA) is considered a standard treatment, but its use is associated with risks, including hemorrhagic complications, and is limited by a transient "therapeutic window" In this regard, the use of nanotechnology to deliver rtPA may be a revolutionary approach. The authors show that nanosystems, especially those produced on the basis of cell membranes (CMD), have the potential to improve the penetration of rtPA into the thrombus and to reduce the infarct volume more significantly. Such delivery methods are thought to help reduce side effects, particularly the risk of bleeding, hemorrhagic transformation and edema. CMD nanomedicine is positioned as an innovative method that offers improved biocompatibility and prolonged circulation time in the blood.
The value of this analysis to readers lies in the fact that it summarizes current advances in nanomedicine for stroke treatment and highlights the prospects for improving standard therapies. This is important for both scientists and practitioners interested in new therapeutic approaches.
However, I have some criticisms that should be addressed to improve the work
1. Lack of clear comparison with clinical rtPA use. The study does not include a direct comparison with conventional rtPA treatment without the use of nanoparticles. This makes it difficult to assess the most important questions:
o To what extent does the use of nanotechnology reduce the negative effects of rtPA, such as bleeding and hemorrhagic transformation?
Is there an actual improvement in therapeutic efficacy, or is this benefit only observed under certain conditions (e.g. in animal models)? Without these data, it is impossible to draw definitive conclusions about the clinical value of nanomedicine.
2. Exclusion criteria. Approximately 700 articles were excluded from the initial search, which represents the vast majority. It is important to clarify:
- What criteria were used to select the studies?
- Were important papers excluded for formal reasons (e.g. language of publication or lack of full text)?
- Did the narrow sample affect the representativeness of the results? Insufficient transparency on this point can affect the credibility of the conclusions.
3. Photothrombosis models disadvantages. The mechanism of thrombus formation in the photothrombosis model (using a laser and photosensitizing agents) differs significantly from pathological thrombosis in patients with ischemic stroke. In particular:
- Such clots have a lower sensitivity to thrombolysis, making the results on the efficacy of rtPA unreliable.
- The study should have focused on this limitation and taken it into account when interpreting the data. The lack of such an analysis may lead to overestimation of the significance of the results. To consider the limitations associated with the use of photothrombosis models and to examine in more detail alternative models that better reflect clinical reality. Compare effects in different models clearly.
Author Response
Decision on the Manuscript ID pharmaceutics-3431099 for Pharmaceutics
Thank you for your consideration of our article in Pharmaceutics journal and for the positive recommendations made by the Reviewers prior to acceptance for publication.
A point-by-point response has been included in the Reply to Reviewers section of the journal portal.
Changes in the MS and SI have been labeled in yellow.
Authors’ response to Reviewer 2 comments
This study is of great importance as ischemic stroke remains one of the leading causes of death and disability worldwide. Recombinant tissue plasminogen activator (rtPA) is considered a standard treatment, but its use is associated with risks, including hemorrhagic complications, and is limited by a transient "therapeutic window" In this regard, the use of nanotechnology to deliver rtPA may be a revolutionary approach. The authors show that nanosystems, especially those produced on the basis of cell membranes (CMD), have the potential to improve the penetration of rtPA into the thrombus and to reduce the infarct volume more significantly. Such delivery methods are thought to help reduce side effects, particularly the risk of bleeding, hemorrhagic transformation and edema. CMD nanomedicine is positioned as an innovative method that offers improved biocompatibility and prolonged circulation time in the blood.
The value of this analysis to readers lies in the fact that it summarizes current advances in nanomedicine for stroke treatment and highlights the prospects for improving standard therapies. This is important for both scientists and practitioners interested in new therapeutic approaches.
However, I have some criticisms that should be addressed to improve the work
We thank the Reviewer for her/his friendly and kind comments.
- Lack of clear comparison with clinical rtPA use. The study does not include a direct comparison with conventional rtPA treatment without the use of nanoparticles. This makes it difficult to assess the most important questions:
We apologize that this critical point was not clearly explained in the first version of the manuscript. The control group used for comparison with the nanoparticles was always the group of animals treated with rtPA. This was chosen because the thrombolytic effect of the administration of the nanosystems was the focus of the analysis.
We have clarified this in line 162-164.
o To what extent does the use of nanotechnology reduce the negative effects of rtPA, such as bleeding and hemorrhagic transformation?
In the case of this meta-analysis, two of the included articles demonstrate reductions in hemorrhagic transformation and bleeding after treatment with rtPA, by monitoring subarachnoid hemorrhage (Biomaterials. 2019 Sep:215:119209) and bleeding time (J Nanobiotechnology. 2024 Jan 3;22(1):10), with reductions between 90 and 70% respectively. Other articles show reductions in various parameters associated with a reduction in hemorrhagic transformation. For example, the study by Yu et al (Acta Biomater. 2022 Mar 1:140:625-640) shows a reduction in free radicals and the study by Quan et al (Adv Healthc Mater. 2022 Aug;11(16):e2200416) shows that coagulation parameters are not altered, in contrast to the rtPA-treated group.
Is there an actual improvement in therapeutic efficacy, or is this benefit only observed under certain conditions (e.g. in animal models)? Without these data, it is impossible to draw definitive conclusions about the clinical value of nanomedicine.
This therapeutic improvement has only been analysed in preclinical models, as the use of nanoparticles with rtPA has not yet been approved in any pathology. Our aim was to analyse preclinical studies in ischemic stroke evaluating different types of nanoparticles for thrombolytic treatment, in order to facilitate future research using this technology that will allow the use of these nanomedicines in clinical practice.
- Exclusion criteria. Approximately 700 articles were excluded from the initial search, which represents the vast majority. It is important to clarify:
- What criteria were used to select the studies?
The inclusion criteria were: (1) preclinical studies of exclusively ischemic stroke; (2) in which rtPA nanoparticles or microparticles were administered. These nanosystems had to include the drug in their structure, that is when rtPA was administered independently of the nanoparticle, it was not included. (3) The infarct volume and/or neurological score had to be reported as outcome measures.
Inclusion criteria are detailed in the Material and methods section, between lines 141-147, which has been modified to clarify this issue.
- Were important papers excluded for formal reasons (e.g. language of publication or lack of full text)?
No articles were excluded because of language or, in this case, missing text. In a first screening, articles were excluded if they were retracted, did not relate to stroke, nanoparticles or rtPA, were in vitro studies or if rtPA was administered as an adjuvant. Articles that were fully screened were excluded because rtPA was administered as an adjuvant or thrombolysis was not performed in the brain.
- Did the narrow sample affect the representativeness of the results? Insufficient transparency on this point can affect the credibility of the conclusions.
In the case of this meta-analysis, we tried to use the most similar preclinical studies that investigated thrombolytic nanoparticles and tested their effect in ischemic stroke models. By using specific inclusion criteria, we tried to reduce possible bias in the selection of articles.
It is true that we only included 18 scientific articles, but it is also important to highlight that the oldest is from 2018, which shows that preclinical studies in this field are relatively recent. In addition, we followed research guidelines, such as PRISMA, to be able to draw conclusions to support and guide future research in this area of study.
- Photothrombosis models disadvantages. The mechanism of thrombus formation in the photothrombosis model (using a laser and photosensitizing agents) differs significantly from pathological thrombosis in patients with ischemic stroke. In particular:
- Such clots have a lower sensitivity to thrombolysis, making the results on the efficacy of rtPA unreliable.
We thank the Reviewer for this note. We fully agree with this comment. Most of the studies are performed using models with platelet-rich thrombus, which may lead to a bias in the results. This is remarked in the discussion, on lines 387-393.
- The study should have focused on this limitation and taken it into account when interpreting the data. The lack of such an analysis may lead to overestimation of the significance of the results. To consider the limitations associated with the use of photothrombosis models and to examine in more detail alternative models that better reflect clinical reality. Compare effects in different models clearly.
The Referee is right. We have performed a further sub-analysis depending on the stroke model used. The studies have been divided into platelet-rich thrombus models (photothrombotic and FeCl3, n=10), fibrin-rich thrombi (thrombin model, n=4) and intraluminal filament model (MCAo, n=6).
A section has been added to the results with this sub-analysis (line 335). Contrary to expectations, the filament and platelet-rich models show the greatest reduction in infarct volume. This may be due to the reduced number of studies, for example, only 4 investigations used the thrombin model, one of which is the study that showed negative results when using nanoparticles. An explanation has been added to the discussion (line 398-406).

Reviewer 3 Report
Comments and Suggestions for Authors
According to the guidelines, the abstract should have around 250 words. Please consider removing unnecessary words. Also, compared to the rest of the text in the abstract, the conclusion in the abstract is short.
Line 61: What about the potential impact of ischemia-reperfusion injury after the administration of rtPA? Reperfusion can induce secondary neuronal injury.
Lines 72-80: Could you briefly write about this nanomedicines approaches, in what way are they different? But just briefly.
Line 110: biocom-patibility
Line 125: Did you consider including other bases?
Line 132: Why were there no restrictions on the language of publication? Could including studies from non-English language journals introduce any potential bias into the results?
Are there any potential safety concerns or limitations associated with CMD nanomedicines that should be addressed in future studies?
Overall, this paper is written appropriately, it is well-structured and interesting to read.
Author Response
Decision on the Manuscript ID pharmaceutics-3431099 for Pharmaceutics
Thank you for your consideration of our article in Pharmaceutics journal and for the positive recommendations made by the Reviewers prior to acceptance for publication.
A point-by-point response has been included in the Reply to Reviewers section of the journal portal.
Changes in the MS and SI have been labeled in yellow.
Authors’ response to Reviewer 3 comments
According to the guidelines, the abstract should have around 250 words. Please consider removing unnecessary words. Also, compared to the rest of the text in the abstract, the conclusion in the abstract is short.
Response: We thank the Reviewer for the indication. We have reduced the length of the abstract.
Line 61: What about the potential impact of ischemia-reperfusion injury after the administration of rtPA? Reperfusion can induce secondary neuronal injury.
Response: We agree with the Reviewer. We have added this side effect of thrombolytic therapy in the lines 69-70.
Lines 72-80: Could you briefly write about this nanomedicines approaches, in what way are they different? But just briefly.
Response: From a physicochemical point of view, the composition and size of these nanosystems are different, however, they have achieved favourable results in encapsulating rtPA, increasing its half-life, increasing its efficacy and decreasing its side effects.
We have added a brief note on this in the introduction (line 73).
Line 110: biocom-patibility
Response: We have corrected this grammatical error.
Line 125: Did you consider including other bases?
Response: We considered other databases but decided to use only PubMed because of the accessibility of relevant articles in medicine, specifically in our case in the field of nanomedicine and ischemic stroke.
Line 132: Why were there no restrictions on the language of publication? Could including studies from non-English language journals introduce any potential bias into the results?
Response: We did not limit the search by language to minimise publication bias and to avoid missing information. Excluding articles written in another language could increase bias in the results. In this case, we did not find any articles in languages other than English.
Are there any potential safety concerns or limitations associated with CMD nanomedicines that should be addressed in future studies?
Response: We thank the Reviewer for his/her comments. Based on the results obtained, we believe that CMD nanomedicines are the safest nanomedicines to use in future studies. As they are mainly composed of cell membranes or compounds found in cell membranes, toxicity problems to different organs are avoided. The only difficulty we might encounter in the future in translating this to humans would be if these nanoparticles were made with human cell membranes, since they would have to be approved by the relevant ethics committees. However, this could be solved by developing nanoplatforms using compounds similar to those found in cell membranes.
Overall, this paper is written appropriately, it is well-structured and interesting to read.
Response: We thank the Reviewer for her/his possitive comments.

Round 2
Reviewer 2 Report
Comments and Suggestions for Authors
All comments and recommendations mentioned in the previous review have been fully taken into account. The changes have significantly improved the structure, content and clarity of presentation of the material. The text has become more logical and consistent, the methodology is described in detail and the results are presented clearly and convincingly. The corrections in wording and terminology have increased the scientific rigor of the work.
I am completely satisfied with the corrected version and believe that the article in its current form fulfills all requirements and can be recommended for publication.